# Data-driven long-term glycaemic control trajectories and their associated health and economic outcomes in Finnish patients with incident type 2 diabetes

Piia Lavikainen[1]*, Emma Aarnio[1], Miika Linna[2], Kari Jalkanen[1], Hilkka Tirkkonen[3], Päivi Rautiainen[3], Tiina Laatikainen[3,4,5], Janne Martikainen[1]

1 School of Pharmacy, University of Eastern Finland, Kuopio, Finland, 2 Aalto University, Helsinki, Finland, 3 Siun Sote – Joint Municipal Authority for North Karelia Social and Health Services, Joensuu, Finland, 4 Department of Public Health Solutions, National Institute for Health and Welfare, Helsinki, Finland, 5 Institute of Public Health and Clinical Nutrition, University of Eastern Finland, Kuopio, Finland

* piia.lavikainen@uef.fi

**Data Availability Statement:** Access to data is regulated by the European Union and Finnish laws and therefore, sharing of sensitive data is not

## Abstract

### Background

Treatments should be customized to patients to improve patients' health outcomes and maximize the treatment benefits. We aimed to identify meaningful data-driven trajectories of incident type 2 diabetes patients with similarities in glycated haemoglobin (HbA1c) patterns since diagnosis and to examine their clinical and economic relevance.

### Materials and methods

A cohort of 1540 patients diagnosed in 2011–2012 was retrieved from electronic health records covering primary and specialized healthcare in the North Karelia region, Finland. EHRs data were compiled with medication purchase data. Average HbA1c levels, use of medications, and incidence of micro- and macrovascular complications and deaths were measured annually for seven years since T2D diagnosis. Trajectories were identified applying latent class growth models. Differences in 4-year cumulative healthcare costs with 95% confidence intervals (CIs) were estimated with non-parametric bootstrapping.

### Results

Four distinct trajectories of HbA1c development during 7 years after T2D diagnosis were extracted: patients with "Stable, adequate" (66.1%), "Slowly deteriorating" (24.3%), and "Rapidly deteriorating" glycaemic control (6.2%) as well as "Late diagnosed" patients (3.4%). During the same period, 2.2 (95% CI 1.9–2.6) deaths per 100 person-years occurred in the "Stable, adequate" trajectory increasing to 3.2 (2.4–4.0) in the "Slowly deteriorating", 4.7 (3.1–6.9) in the "Rapidly deteriorating" and 5.2 (2.9–8.7) in the "Late diagnosed" trajectory. Similarly, 3.5 (95% CI 3.0–4.0) micro- and macrovascular complications per 100 person-years occurred in the "Stable, adequate" trajectory increasing to 5.1 (4.1–6.2) in the "Slowly deteriorating", 5.5 (3.6–8.1) in the "Rapidly deteriorating" and 7.3 (4.3–11.8) in the

possible and data are not publicly available. An anonymised version of the data is available for researchers who meet the criteria as required by the European Union and Finnish laws for access to confidential data with a data permit of an appropriate authority. Contact information: aineistoneuvonta@siunsote.fi for requests from the Siun sote—Joint municipal authority for North Karelia social and health services and tietoaineistot@kela.fi for data requests from the Social Insurance Institute.

**Funding:** This study was partly supported by the Social Insurance Institution of Finland (https://www.kela.fi/web/en, diary number 110/522/2018), Finnish Diabetes Association (https://www.diabetes.fi/en/finnish_diabetes_association), the Research Committee of the Kuopio University Hospital Catchment Area for the State Research Funding (https://www.psshp.fi/web/en, project QCARE, Joensuu, Finland), and the Strategic Research Council at the Academy of Finland (https://www.aka.fi/en/strategic-research/, project IMPRO, 312703). The funders had no role in the design of the study and collection, analysis, and interpretation of data and in writing the manuscript. The specific roles of these authors are articulated in the 'author contributions' section.

**Competing interests:** I have read the journal's policy and the authors of this manuscript have the following competing interests: JM is a founding partner of ESiOR Oy and a board member of Siltana Oy. EA has received consulting and lecturing fees from Merck & Co and UCB Pharma. These companies were not involved in carrying out this research. This does not alter our adherence to PLOS ONE policies on sharing data and materials. PL, ML, KJ, HT, PR and TL declare no conflicts of interest.

"Late diagnosed" trajectory. Patients in the "Stable, adequate" trajectory had lower accumulated 4-year medication costs than other patients.

## Conclusions

Data-driven patient trajectories have clinical and economic relevance and could be utilized as a step towards personalized medicine instead of the common "one-fits-for-all" treatment practices.

## Introduction

Main target of type 2 diabetes (T2D) care is to achieve and maintain good treatment balance to avoid complications and premature deaths [1–4]. Typically, glycated haemoglobin A1c (HbA1c) is used as a measure of long-term blood sugar level with values <53 mmol/mol (7.0%) indicating good treatment balance. However, individualized, more-stringent goals for young T2D patients with short disease duration and less-rigorous, personalized treatment targets for older and the most comorbid patients are recommended [1, 4]. Treatments should be customized to patients to improve patients' health outcomes and maximize the treatment benefits.

While the need of individual tailoring of treatments due to heterogeneity of T2D patients is recognized, majority of patients still receive standardized "*one-size-fits-all*" care due to limitations in available resources for individualized treatment. However, identification of patient groups with similarities in care needs and treatment outcomes, taking also into account budget constraints and cost consequences, is a step towards personalized care. It is also economically rational to identify subgroups where the treatment is cost-effective [5] and to offer care accordingly. Clustering of treatment outcomes, namely HbA1c, of T2D patients provides tools for treatment targeting and service design, and, further, optimizing the allocation of limited economic resources. Patients in different trajectories are likely to prefer different configurations of diabetes care and support. For example, patients with a deteriorating treatment balance profile should receive more intensive disease management supported by health professionals whereas adequate self-management support could be well enough for patients within stable treatment balance.

The objective of this study was to identify meaningful data-driven trajectories of incident T2D patients with similarities in HbA1c patterns since diagnosis. Further, we aimed to describe incidence of diabetes complications, all-cause mortality, and accumulation of healthcare costs by the trajectories to validate clinical applicability of the results in optimization and supporting effectiveness of treatment.

## Materials and methods

### Study setting

In Finland, most health services are public meaning all residents have the right to use and equal access to them. Municipalities are responsible for organizing and financing public health and social care services for their residents. Siun sote—Joint municipal authority for North Karelia social and health services was established in the region of North Karelia at the beginning of 2017 [6]. Instead of one municipality, Siun sote joins healthcare organisations of 13 municipalities with around 166 000 inhabitants. Before the integration to Siun sote, all the

municipalities in the North Karelia region had introduced an identical electronic health regis-
ter (EHR) called Mediatri in 2010–2011, and its use has been continued also after the integra-
tion. EHR data from the Mediatri cover both public primary and specialized healthcare.
However, these EHRs do not cover patients utilizing only private healthcare services that pro-
duce about one quarter of all the health and social care services.

All community-dwelling residents of Finland are eligible for medication reimbursement,
and reimbursed medication purchases from community pharmacies are registered in the Finn-
ish Prescription Register (FPR) established in 1994 and maintained by the Social Insurance
Institution of Finland (SII). However, patients staying at public nursing homes or hospitals
>90 consecutive days are not eligible for medication reimbursement and their purchases are
not registered during these stays. In addition, the FPR does not include records of over-the-
counter medication purchases. The Special Reimbursement Register (SRR), established in
1964 and also maintained by the SII, includes records of entitlements to higher medication
reimbursements due to certain chronic diseases, such as T2D.

## Study design

This study utilized data provided by Siun sote, where the prevalence of T2D was 6.2% in 2012
[7]. All patients with newly diagnosed T2D in 2011–2012 were identified with ICD-10 code [8]
E11 from the EHRs. Collected EHR data contained information on diagnoses, medication pre-
scriptions, as well as laboratory assays during 2010–2017. EHR data were combined with infor-
mation on medication purchases (such as dispensing date, Anatomical Therapeutic Chemical
(ATC) classification code [9], and cost) from the FPR for the years 1995–2017. Entitlements to
special refund for diabetes medications prior to 2011 were retrieved from the SRR.

Timing of T2D diagnosis was ascertained with data on diabetes medication purchases
(ATC code A10) from the FPR in 1995–2010 and the retrieved entitlements to special refund
for diabetes medications since 1964. If a patient had T2D diagnosed by healthcare providers
outside of Siun sote, for example, before moving to the North Karelia region, the first record of
T2D could be tracked from the national FPR and SRR instead of the first record in the EHRs
of Siun sote. T2D diagnosis date was considered to be the first ever record of either diabetes
medication purchase in the FPR, entitlement to special refund in the SRR, or T2D diagnosis in
the EHRs, and only patients with an ascertained diagnosis recorded in 2011–2012 were
included.

## Glycaemic control

Glycaemic control was measured with HbA1c with the turbidimetric inhibition immunoassay
method (TINIA) in the Eastern Finland laboratory (ISLAB, https://www.islab.fi). The method
and its analysis were standardized across the North Karelia region. Values were standardized
to the International Federation of Clinical Chemistry (IFCC) units. EHRs data on measure-
ments from 3 months prior to T2D diagnosis until a maximum of 89 months after it were
used. Patients with ≥1 HbA1c measurement during the study period were included. Baseline
HbA1c (mmol/mol) level was defined to be the one measured ±3 months from T2D diagnosis.
Follow-up measures were categorized as occurring 6–17 (1 year), 18–29 (2 years), 30–41 (3
years), 42–53 (4 years), 54–65 (5 years), 66–77 (6 years), and 78–89 months (7 years) after diag-
nosis. If several measurements occurred within a period, the latest one was selected. Sensitivity
analyses were conducted against the average of the HbA1c measurements within a period and
the results were comparable. If HbA1c value was unmeasured within a certain period, it was
considered as a missing value. HbA1c values were measured until the date of death, moving
outside of the study area, or 31st December 2017, whichever occurred first.

## Deaths and diabetes complications

Information on the dates of deaths and complications occurring after T2D diagnosis were retrieved from the EHRs. Micro- and macrovascular complications were identified with ICD-10 codes which have been in use in Finland since 1996 (S1 Table). Complications were examined as a composite of micro- and macrovascular complications as well as separately. Incidence of all-cause deaths and diabetes complications were followed up until the date of the specific event, moving outside of the study area, or 31st December 2017, whichever occurred first.

## Healthcare costs

Healthcare costs were calculated as a cumulative sum of primary and specialized healthcare costs, nursing home and home care costs, and outpatient medication costs during 2014–2017. EHR data included all hospitalizations (outpatient visits, inpatient admissions) and their costs derived from the hospitals' cost accounting systems in 2014–2017. Restriction in cost analyses to years 2014–2017 was applied because data on outpatient visits were not complete and, therefore, comparable between municipalities during the transitional phase to the Mediatri before 2014. The individual-level resource use data for primary care were also obtained from the EHRs (with diagnosis and activity information) and grouped using the Ambulatory and Primary Care-Related Patient Groups (APR) grouper software, a grouping system equivalent to DRG used in hospital care. The unit cost estimates for grouped primary care contacts were obtained from the national standard price lists for primary care encounters [10]. Outpatient medication costs were calculated based on the information on the total costs of all reimbursed as well as reimbursed antidiabetic (ATC code A10) medications only from the FPR. In addition to the combined total costs, accumulated social- and healthcare costs and medication costs were examined separately. Costs were estimated from the payer's perspective, including all direct health and social care costs.

## Baseline characteristics

Patients' age and gender were obtained from the EHRs. Concordant and discordant diseases that occurred prior to the T2D diagnosis date were defined from the EHRs (S2 Table). Micro- and macrovascular comorbidities occurring prior to T2D diagnosis were identified utilizing the same definitions as for the outcome (S1 Table). Statin (ATC code C10A) use was identified from the FPR.

For body mass index (BMI), blood pressure, fasting plasma glucose (FPG), and creatinine, measurements occurring most closely to the T2D diagnosis date measured before or within 3 months from the T2D diagnosis, and for low-density lipoprotein (LDL), measurements occurring most closely to the T2D diagnosis date measured before or within 1 month from the diagnosis were extracted from the EHRs. Based on these definitions, indicators of having LDL, BMI, FPG, blood pressure, creatinine and HbA1c measured at the baseline were created. In addition, estimated glomerular filtration rate (eGFR) was calculated with the Chronic Kidney Disease Epidemiology Collaboration (CKD-EPI) equation [11] based on creatinine, age, and gender. eGFR was further classified into normal/high (stage I, $>90$ ml/min/1.73 m$^2$), mildly decreased (stage II, 60–90 ml/min/1.73 m$^2$), mildly to severely decreased (stage III, 30–59 ml/min/1.73 m$^2$) and severely decreased/kidney failure (stage IV, $<30$ ml/min/1.73 m$^2$).

## Diabetes medication use during the follow-up

Diabetes medication use was examined from the FPR for 7 years since T2D diagnosis. The following subgroups of medication use were formed: users of metformin (ATC code A10BA02)

only, users of metformin and other oral antidiabetic medications (A10BA02 + other A10B), users of only other diabetes medications than insulin or metformin (A10B excluding A10BA02 and A10A), users of insulin and oral antidiabetic medication (including metformin) (A10A + A10B), and users of insulin (A10A) only. Patients who died were taken into account in the calculation of prevalence of diabetes medication use.

## Statistical analyses

Patients were clustered into distinct trajectories of glycaemic control with latent class growth analysis (LCGA) [12–15]. Models were fitted iteratively with 1–5 classes and varying shapes for the trajectories (linear, quadratic, cubic). LCGAs were estimated with full-information maximum likelihood and missing data were not imputed but all available data were used. The method assumes no variation within the trajectories. Selection of the best model was done utilizing information from fit indices and classification performance of the model as well as clinical interpretation of the trajectories. Bayesian Information Criteria (BIC) was utilized to compare models with lower values indicating better fit. Low-Mendel-Rubin likelihood ratio test with P value <0.05 was considered to indicate better fit for a $n$-class model than for $n$–1-class model. Entropy was used to guide in the classification accuracy of the model with higher values indicating better classification. Small classes with <2% of population were not accepted to have large enough groups of at least 30 patients. LCGAs were performed using Mplus [16] Version 8.

Differences in baseline characteristics between the identified trajectories were examined with one-way ANOVA or nonparametric Kruskal-Wallis test for continuous variables and chi squared test for categorical ones. Association of baseline characteristics with trajectory membership was examined with multinomial logistic regression model. Cox regression models were utilized to describe incidence of deaths and micro- and macrovascular complications by the trajectories. Models were adjusted for age, sex, concordant and discordant diseases, and microvascular and macrovascular comorbidities at baseline. Calendar year- and trajectory-specific healthcare costs were weighted with Kaplan-Meier sample average estimator [17] to account for differences between trajectories in censoring due to death in calculation of adjusted cumulative costs during 2014–2017. Further, non-parametric bootstrapping generating 1000 bootstrap samples with replacement was applied to estimate mean differences in adjusted cumulative healthcare costs with 95% confidence intervals (CIs). Analyses were conducted with SAS version 9.4 (SAS Institute Inc., Cary, North-Carolina, USA). Two-tailed P values < 0.05 were considered statistically significant.

## Ethics statement

Use of the data was approved by the Ethics Committee of the Northern Savonia Hospital District (diary number 81/2012) that considers all applications within its university hospital catchment area. The study protocol was also approved by the register administrator, Siun sote—Joint municipal authority for North Karelia social and health services. A separate permission to link data on medication purchases and special reimbursements was achieved from the Social Insurance Institute (diary number 110/522/2018). We utilized only de-identified register-based data and individuals in the registers were not contacted. Thus, no written consent from the patients was required.

## Results

In total, 1647 T2D patients diagnosed in 2011–2012 were identified. After removing patients without HbA1c measurements during the follow-up, 1540 (93.5%) patients remained with

**Table 1. Baseline characteristics and variables describing treatment monitoring during the follow-up overall and by the estimated HbA1c trajectories.**

| | Overall cohort | Stable, adequate glycaemic control | Slowly deteriorating glycaemic control | Rapidly deteriorating glycaemic control | Late diagnosed patients | P value |
|---|---|---|---|---|---|---|
| n (%) | 1540 (100.0) | 1058 (68.7) | 340 (22.1) | 95 (6.2) | 47 (3.1) | |
| **Baseline characteristics** | | | | | | |
| Female, n (%) | 716 (46.5) | 514 (48.6) | 154 (45.3) | 32 (33.7) | 16 (34.0) | 0.010 |
| Mean age (SD), years | 64.9 (12.7) | 65.3 (12.2) | 65.5 (13.4) | 61.1 (14.2) | 60.9 (14.2) | 0.002 |
| FPG measured, n (%) | 1122 (72.9) | 782 (73.9) | 246 (72.4) | 73 (76.8) | 21 (44.7) | <0.001 |
| Mean FPG (SD), mmol/l | 7.6 (2.3) | 7.0 (1.4) | 8.2 (2.6) | 10.5 (4.1) | 12.6 (4.1) | <0.001 |
| HbA1c measured, n (%) | 935 (60.7) | 608 (57.5) | 234 (68.8) | 60 (63.2) | 33 (70.2) | 0.001 |
| Mean HbA1c (SD), mmol/mol | 48.5 (16.5) | 40.8 (4.8) | 47.6 (6.8) | 56.0 (11.0) | 95.2 (15.1) | <0.001 |
| Blood pressure measured, n (%) | 526 (34.2) | 382 (36.1) | 108 (31.8) | 29 (30.5) | 7 (14.9) | 0.012 |
| LDL measured, n (%) | 1016 (66.0) | 721 (68.2) | 207 (60.1) | 69 (72.6) | 19 (40.4) | <0.001 |
| Mean LDL (SD), mmol/l | 3.0 (1.0) | 3.0 (0.9) | 3.0 (1.0) | 3.3 (1.0) | 3.5 (1.3) | 0.008 |
| HbA1c, LDL and blood pressure measured, n (%) | 302 (19.6) | 210 (19.9) | 70 (20.6) | 18 (19.0) | 4 (8.5) | 0.268 |
| Creatinine measured, n (%) | 1286 (83.5) | 889 (84.0) | 282 (82.9) | 79 (83.2) | 36 (76.6) | 0.588 |
| Mean creatinine (SD), μmol/l | 76.7 (35.3) | 76.9 (39.2) | 76.5 (21.9) | 79.9 (32.8) | 66.4 (21.2) | 0.289 |
| Classified eGFR (ml/min/1.73 m$^2$) | | | | | | |
| Stage IV (<30) | 8 (0.6) | 7 (0.8) | 1 (0.4) | 0 | 0 | NA |
| Stage III (30–59) | 120 (9.3) | 73 (8.2) | 31 (11.0) | 11 (13.9) | 5 (13.9) | 0.219 |
| Stage II (60–90) | 518 (40.3) | 369 (41.5) | 116 (41.1) | 26 (32.9) | 7 (19.4) | 0.020 |
| Stage I (>90) | 640 (49.8) | 440 (49.5) | 134 (47.5) | 42 (53.2) | 24 (66.7) | 0.446 |
| BMI measured, n (%) | 673 (43.7) | 487 (46.0) | 144 (42.4) | 33 (34.7) | 9 (19.1) | 0.001 |
| Mean BMI (SD), kg/m$^2$ | 31.0 (6.1) | 30.8 (5.9) | 31.3 (6.6) | 33.3 (6.3) | 28.8 (5.9) | 0.083 |
| Concordant diseases, n (%) | 473 (30.7) | 337 (31.9) | 108 (31.8) | 24 (25.3) | 4 (8.5) | 0.005 |
| Discordant diseases, n (%) | 202 (13.1) | 149 (14.1) | 36 (10.6) | 12 (12.6) | 5 (10.6) | 0.384 |
| Prevalence of vascular comorbidities, n (%) | 158 (10.3) | 108 (10.2) | 41 (12.1) | 6 (6.3) | 3 (6.4) | 0.312 |
| Microvascular, n (%) | 39 (2.5) | 30 (2.8) | 7 (2.1) | 2 (2.1) | 0 | 0.574 |
| Macrovascular, n (%) | 132 (8.6) | 87 (8.2) | 38 (11.2) | 4 (4.2) | 3 (6.4) | 0.127 |
| Prevalence of vascular comorbidities and concordant diseases, n (%) | 475 (30.8) | 339 (32.0) | 108 (31.8) | 24 (25.3) | 4 (8.5) | 0.004 |
| Statins, n (%) | 646 (41.9) | 472 (44.6) | 125 (36.8) | 39 (41.1) | 10 (21.3) | 0.002 |
| **Variables describing treatment monitoring during the follow-up[a]** | | | | | | |
| No. of HbA1c measurements, mean (SD) | 8.1 (5.2) | 7.2 (4.6) | 10.4 (5.5) | 10.0 (6.5) | 8.2 (6.2) | <0.001 |
| Mean HbA1c (SD) | 46.3 (11.6) | 41.0 (5.5) | 51.0 (9.7) | 65.2 (16.8) | 60.0 (19.7) | <0.001 |

BMI, body mass index; eGFR, estimated glomerular filtration rate; FPG, fasting plasma glucose; LDL, low-density lipoprotein.

[a]Mean follow-up 2016 days (5.5 years); NA, not available.

14,173 HbA1c measurements. Mean (SD) age of the patients was 64.9 (12.7) years and 46.5% were female (Table 1). 935 (60.7%) had HbA1c measured ±3 months of T2D diagnosis with an average HbA1c of 48.5 (16.5) mmol/mol.

Of the fitted LCGAs (S3 Table), four-class cubic model was selected as the best fitting with clinically interpretable results (Fig 1). The extracted trajectories were: patients with "Stable, adequate" (66.1%), "Slowly deteriorating" (24.3%), and "Rapidly deteriorating" glycaemic control (6.2%) as well as "Late diagnosed" patients (3.4%).

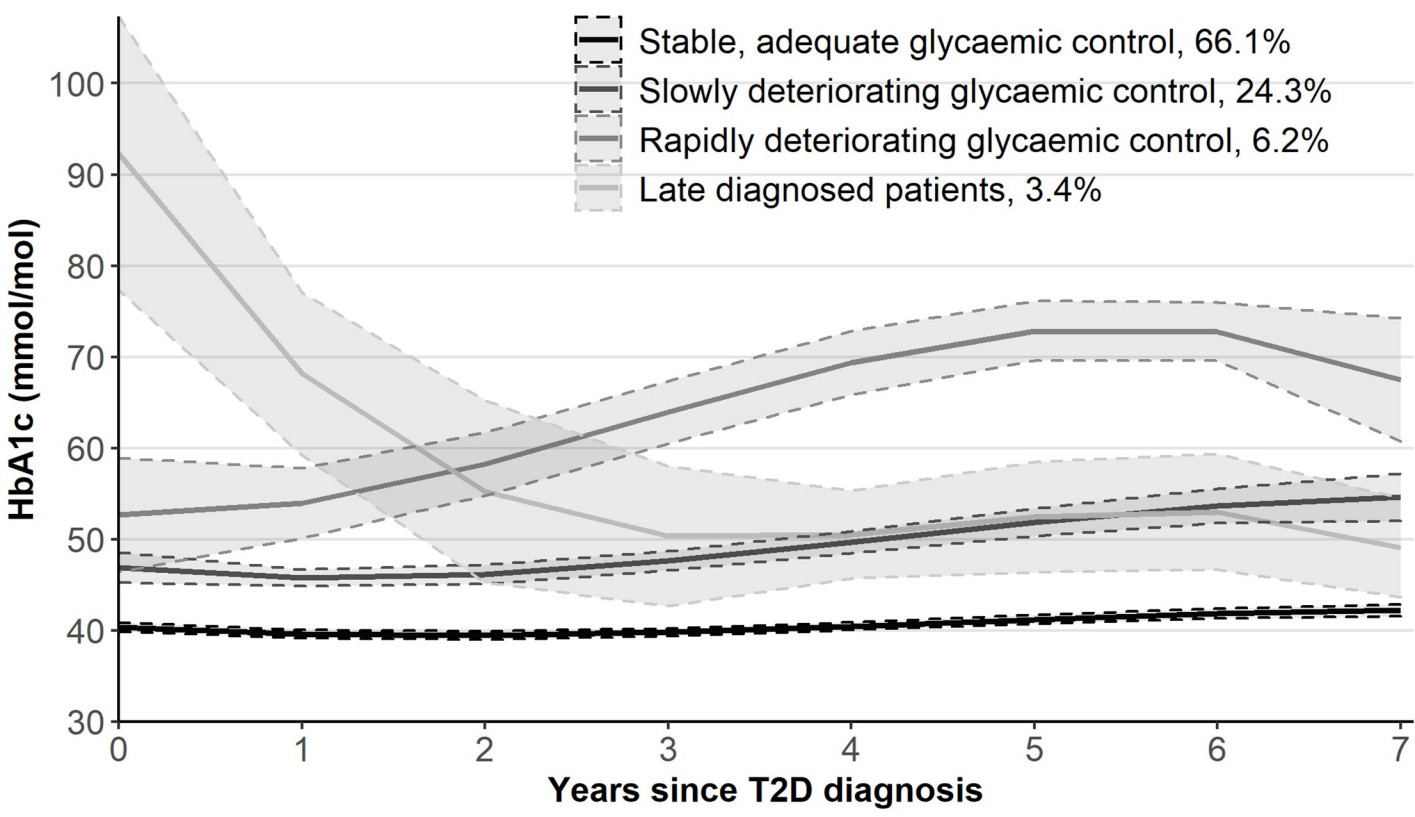

**Fig 1. Estimated HbA1c trajectories.**

Patients in the"Rapidly deteriorating" trajectory were more often male and younger than patients in the"Stable, adequate" trajectory (Table 2). Having HbA1c measured at the time of the T2D diagnosis increased and having LDL measured decreased the odds of belonging to the "Slowly deteriorating" and "Late diagnosed" trajectories compared with the"Stable, adequate" trajectory. Having BMI measured decreased the odds of belonging to the "Rapidly deteriorating" and "Late diagnosed" trajectories compared with the"Stable, adequate" trajectory. Patients in the"Late diagnosed" trajectory had less concordant diseases and patients in the "Slowly deteriorating" trajectory used less statins than patients in the"Stable, adequate" trajectory. Average number of HbA1c measurements during the study period differed between the trajectories (Table 1).

Diabetes medication use differed between the trajectories and the content of medication structure changed during the 5-year follow-up (Fig 2, S4 Table). In the "Stable, adequate" trajectory, half of the patients did not purchase diabetes medications during the follow-up and metformin use was most frequent. Among patients in the "Rapidly deteriorating" trajectory, treatments were clearly intensified throughout the follow-up. Use of insulin was the most frequent in the "Late diagnosed" trajectory with almost half of the patients using it, remaining stable over the follow-up. Non-use of diabetes medications decreased in every trajectory during the follow-up.

In total, 226 deaths and 318 complications (125 microvascular and 237 macrovascular events) occurred during the follow-up (Fig 3, S1 and S5 Tables). Incidence rate of death after T2D diagnosis was 2.2 (95% CI 1.9–2.6) events per 100 person-years among "Stable, adequate" trajectory, increasing to 3.2 (2.4–4.0) in "Slowly deteriorating", 4.7 (3.1–6.9) in "Rapidly

**Table 2. Adjusted, multivariate odds ratios (95% confidence intervals) for the association of baseline characteristics with estimated trajectories (n = 1540).** Stable, adequate glycaemic control trajectory (n = 1058) as a reference.

|  | Slowly deteriorating glycaemic control (n = 340) | Rapidly deteriorating glycaemic control (n = 95) | Late diagnosed patients (n = 47) |
|---|---|---|---|
|  | OR (95% CI) | OR (95% CI) | OR (95% CI) |
| Female | 0.82 (0.64–1.06) | 0.55 (0.35–0.87) | 0.51 (0.27–0.98) |
| Age, years |  |  |  |
| <50 | 1.00 | 1.00 | 1.00 |
| 50–74 | 0.74 (0.49–1.12) | 0.37 (0.21–0.64) | 0.48 (0.22–1.06) |
| ≥75 | 1.08 (0.68–1.73) | 0.37 (0.18–0.75) | 0.55 (0.20–1.52) |
| HbA1c measured | 1.97 (1.47–2.64) | 1.38 (0.85–2.24) | 3.42 (1.59–7.36) |
| LDL measured | 0.68 (0.51–0.91) | 1.46 (0.84–2.55) | 0.36 (0.19–0.72) |
| Blood pressure measured | 0.84 (0.62–1.13) | 0.99 (0.59–1.66) | 0.52 (0.22–1.25) |
| Creatinine measured | 0.90 (0.61–1.34) | 0.90 (0.46–1.77) | 1.02 (0.44–2.41) |
| BMI measured | 0.95 (0.71–1.27) | 0.54 (0.33–0.90) | 0.39 (0.17–0.86) |
| Statins | 0.75 (0.58–0.98) | 0.91 (0.58–1.43) | 0.51 (0.24–1.07) |
| Concordant diseases or micro- or macrovascular comorbidities | 1.07 (0.81–1.41) | 0.84 (0.50–1.39) | 0.26 (0.09–0.74) |
| Discordant diseases | 0.75 (0.50–1.12) | 1.05 (0.54–2.01) | 1.06 (0.39–2.87) |

BMI, body mass index; CI, confidence interval; LDL, low-density lipoprotein; OR, odds ratio.

deteriorating" and 5.2 (2.9–8.7) in the "Late diagnosed" trajectories. Hazard ratio for death was 2.86 (95% CI 1.60–5.12) in the "Late diagnosed" and 2.49 (95% CI 1.61–3.87) in the "Rapidly deteriorating" trajectory compared with the "Stable, adequate" trajectory (S6 Table). Similarly, incidence rate of micro- and macrovascular complications after T2D diagnosis was 3.5 (95% CI 3.0–4.0) events per 100 person-years among "Stable, adequate" trajectory, increasing to 5.1 (4.1–6.2) in "Slowly deteriorating", 5.5 (3.6–8.1) in "Rapidly deteriorating" and 7.3 (4.3–11.8) in the "Late diagnosed" trajectories. Hazard ratio for complications was 3.35 (95% CI 1.96–5.72) in the "Late diagnosed", 2.28 (95% CI 1.51–3.46) in the "Rapidly deteriorating", and 1.39 (95% CI 1.08–1.79) in the "Slowly deteriorating" compared with the "Stable, adequate" trajectory.

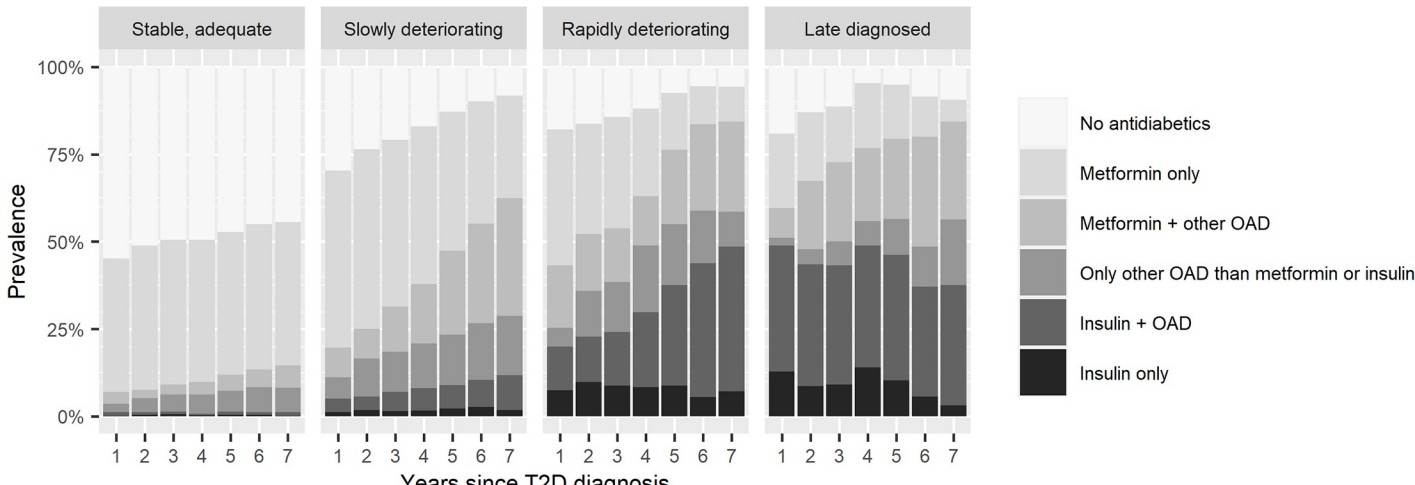

**Fig 2. Prevalence of diabetes medication use by HbA1c trajectories.** OAD, oral antidiabetic drugs or GLP-1 analogues (incl. metformin, sulfonylureas, combinations of oral blood glucose lowering drugs, glitazones, DPP-4 inhibitors, glinides, GLP-1 analogues, and SGLT2 inhibitors).

**a)**

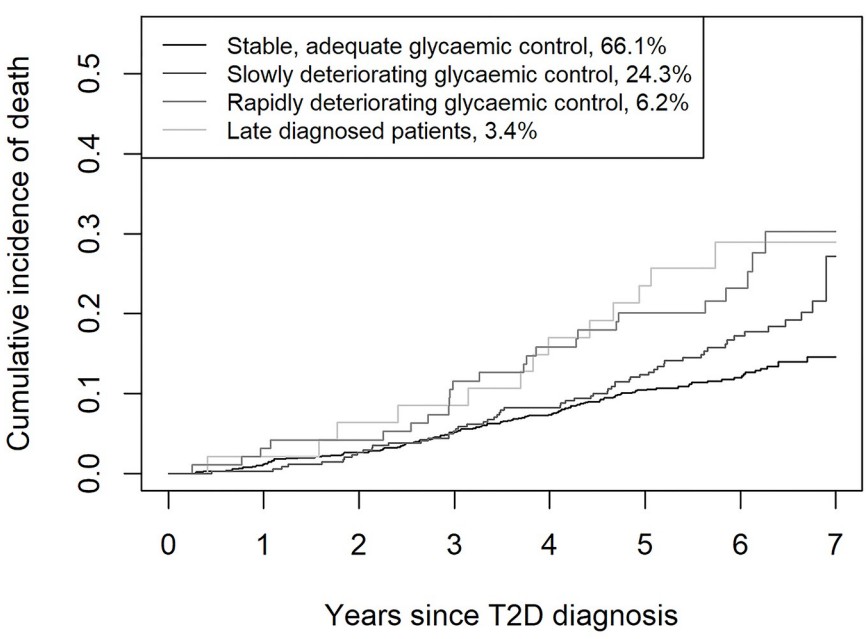

**b)**

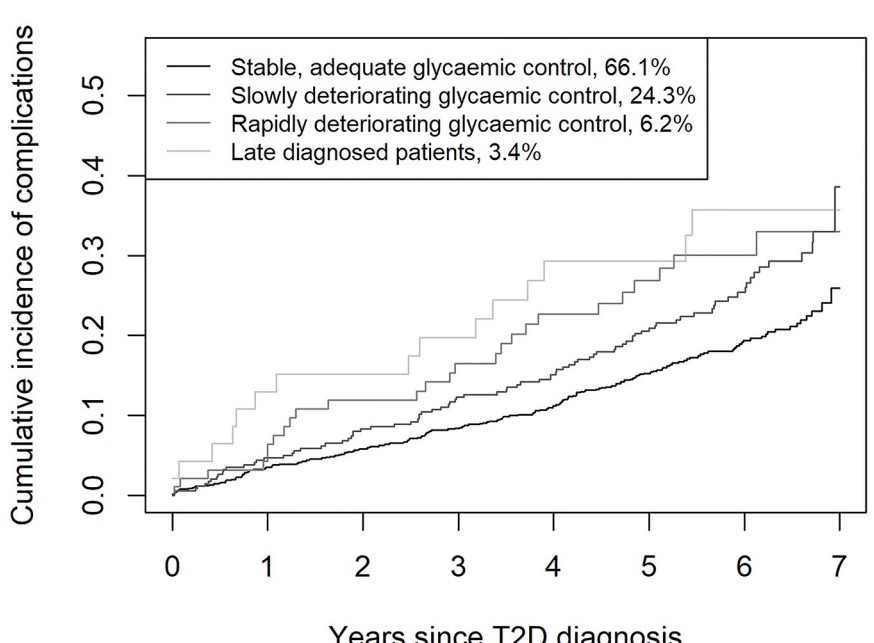

**Fig 3. Cumulative incidences of health outcomes by the four HbA1c trajectories.** (A) Cumulative incidence of death since T2D diagnosis. (B) Cumulative incidence of combined event of micro- and macrovascular complications since T2D diagnosis.

**Table 3. Absolute accumulated 4-year per-person healthcare costs (euros, €) for the Stable, adequate glycaemic control trajectory.** Mean differences in accumulated 4-year per-person healthcare costs between HbA1c trajectories with bootstrapped 95% confidence intervals (Stable, adequate glycaemic control trajectory as a reference). All costs are Kaplan-Meier sample average adjusted and calculated for 2014–2017.

| Trajectory | Accumulated 4-year per-person costs, € | Mean differences in accumulated 4-year per-person costs (€) compared with Stable, adequate glycaemic control trajectory (bootstrapped 95% CI) | | |
|---|---|---|---|---|
| | Stable, adequate glycaemic control | Slowly deteriorating glycaemic control | Rapidly deteriorating glycaemic control | Late diagnosed patients |
| n (%), alive on Jan 1, 2014 | 1032 (68.7) | 335 (22.1) | 90 (6.2) | 44 (3.1) |
| Accumulated total costs | 15 341 | 3882 (-899 to 8872) | 1021 (-4122 to 6522) | 2817 (-7416 to 18 669) |
| 1. Accumulated social- and healthcare costs | 12 124 | 3192 (-1306 to 8015) | -107 (-4749 to 5156) | 1379 (-8443 to 17 069) |
| i) Primary healthcare costs | 3006 | 599 (-423 to 1847) | 478 (-952 to 2188) | -534 (-1843 to 1432) |
| -Outpatient visits | 1659 | 54 (-187 to 295) | 79 (-446 to 696) | -637 (-949 to -302) |
| -Inpatient visits | 1347 | 546 (-421 to 1739) | 398 (-734 to 1903) | 103 (-1094 to 1910) |
| ii) Specialized healthcare costs | 4543 | 793 (-1110 to 3184) | 48 (-2023 to 2522) | -1550 (-3328 to 411) |
| -Outpatient visits | 1920 | -98 (-762 to 551) | -435 (-1134 to 312) | -1035 (-1664 to -418) |
| -Inpatient visits | 2622 | 891 (-679 to 3016) | 483 (-1246 to 2745) | -515 (-1943 to 1094) |
| iii) Nursing home care costs | 1200 | 1767 (-106 to 4035) | -396 (-1585 to 1055) | -750 (-1691 to 456) |
| iv) Home care costs | 3376 | 32 (-2500 to 2629) | -236 (-3354 to 3446) | 4213 (-3541 to 17 773) |
| 2. Accumulated medication costs | 3216 | 690 (2 to 1486) | 1128 (178 to 2211) | 1437 (162 to 2805) |
| i) Accumulated diabetes medication costs | 363 | 795 (649 to 959) | 2081 (1542 to 2657) | 2046 (1283 to 2884) |

Accumulated total medication costs differed between the trajectories while no difference was seen in accumulated total costs or accumulated social- and healthcare costs (Table 3, S6 Table). However, patients in the "Late diagnosed" trajectory had lower outpatient visit costs in both primary and specialized healthcare than patients in the "Stable, adequate" trajectory. In addition, patients in the "Stable, adequate" trajectory had lower accumulated diabetes medication costs than patients in other trajectories.

## Discussion

Based our study results, four trajectories among incident Finnish T2D patients were identified with data-driven technique to classify patients following similar HbA1c patterns over time. Identified trajectories had clinical relevance in terms of diabetes medication use, incidence of diabetes complications and deaths as well as accumulation of medication costs during the study. Monitoring of HbA1c and LDL at the time of T2D diagnosis differed between the trajectories. To our knowledge, novelty of the present study is in the examination of direct long-term costs of social- and healthcare by the trajectories among incident T2D patients in Finland. Due to the restricted healthcare resources, it is crucial to understand the costs associated with the trajectories to give reasons for the most effective care of T2D to avoid costly complications.

Trajectories estimated in this study resemble those identified in previous publications. Typically, 2–5 trajectories have been identified among incident T2D populations [18–21]. More studies have been conducted among prevalent T2D patients [22–27]. Generally, comparison with these previous studies is complicated because of differences in statistical methods used to identify trajectories, follow-up times as well as calendar times: new medications have entered the markets between the studies intensifying the treatment of T2D. In addition, as care and

treatment of diabetes varies between geographical areas, and even within one country depending on healthcare providers in addition to differences between populations, results may not be universally generalizable. To note, also clustering based on patient characteristics (age, HbA1c, BMI, homeostasis model assessment (HOMA2) estimates of β-cell function and insulin resistance) at the time of diagnosis in general diabetic population has been examined [28] instead of longitudinal clustering based on HbA1c solely.

Patients in the "Late diagnosed" trajectory had less concordant diseases and measurements of LDL than patients in the "Stable, adequate" trajectory at the time of T2D diagnosis. In addition, having HbA1c measured closely to the T2D diagnosis date increased the odds to be late diagnosed T2D patient compared with the "Stable, adequate" trajectory. These indicate that T2D diagnosis may be delayed due to nonexistence of other diseases or healthcare problems requiring regular controls. In the previous studies [18–21], younger age, male sex, and higher BMI have been found to be associated with poorer glycaemic control trajectories similarly to our study.

To examine clinical relevance of the identified trajectories, we described incidences of all-cause deaths and diabetes complications as well as accumulation of healthcare costs by the trajectories. Our results are in line with a systematic review [29] of studies utilizing longitudinal HbA1c clustering and examining their associations with outcomes reporting poor glycaemic control to be associated with the risk of micro- and macrovascular events and mortality. HbA1c may not reflect only treatment response but also the underlying severity of the disease [28]. Late diagnosed patients used most often insulin reflecting more severe disease, seen also as higher incidence of complications during the 7-year follow-up despite of reaching good glycaemic control 2–3 years after the diagnosis. We observed that accumulated medication costs differed between the trajectories and poorer glycaemic control was associated with increased costs which could be considered as a natural consequence of more severe disease, treatment intensification and incidence of complications.

Medical treatments have a central role in T2D care. Newer diabetes medications, i.e., SGLT2 inhibitors and GLP-1 receptor agonists, were reported to lower the risk of cardiovascular events and subsequent mortality during the 2010's [3, 4]. Use of these medications, especially SGLT2 inhibitors, started to rise in 2016 in Finland. In our cohort, these medications were most often used by patients from the "Rapidly deteriorating" trajectory (S4 Table). In addition, increased use was observed within the "Late diagnosed" and "Slowly deteriorating" trajectories. When examining the risk of micro- and macrovascular complications and mortality, it seems that our analyses may, therefore, underestimate the underlying risk caused by poorer glycaemic control when compared with the "Stable, adequate" trajectory. Of the complications studied in this study, chronic kidney disease may have been somewhat underdiagnosed leading to underestimation of microvascular diseases at baseline and during the follow-up. As the disease is highly related to the mortality risk, residual confounding in mortality analyses is possible although the analyses accounted for end-stage renal disease.

Strengths of our study are inclusion of all patients with T2D diagnosed in 2011–2012 in the North Karelia region. In addition, all the municipalities in North Karelia use the same regional laboratory and the same standardized methods for HbA1c testing. Due to the Finnish public healthcare system available for all inhabitants, we were able to continuously follow patients' public healthcare service use until death, moving outside of the study area, or administrative study end of the study. Utilization of register-based data allowed us to avoid recall bias, too. Validation studies of the national Finnish Care Register for Health Care, which also includes EHRs from the North Karelia region, have reported positive predictive values of 87–97% for hospital discharges of vascular diseases [30]. Compared with the previous studies, we were also able to examine economic outcomes in addition to health outcomes.

Our study has some limitations. We had a relatively small sample of incident T2D patients stratified to trajectories which limits the conclusions we can draw from the results due to wide confidence intervals especially in the smallest trajectories, although point estimates show some interesting tendencies. For this, we were also unable to conduct an internal validation study. Although we ascertained the timing of T2D diagnosis observed in the EHRs with data from the SII, the date of diagnosis may not be accurate. We had scarce data on laboratory measurements at the baseline meaning that we could not examine the effect of some important factors, such as LDL or BMI, as such on trajectory probability. In addition, according to our previous study, younger patients with T2D were poorly monitored compared with older patients in the North Karelia region [31]. We did not incorporate information on the diabetic medications in use to our analyses but focused solely on HbA1c levels and associated outcomes. However, differences are reported between medications in their efficacy to manage HbA1c [3, 4].

Interpretation of costs should be cautious. Cumulative healthcare costs did not include costs from the first 1–2 years of follow-up since diagnosis. However, as depicted from Fig 3B, differences in the incidence of micro- and macrovascular complications between the trajectories were already apparent at that time possibly causing some bias, especially to specialized healthcare cost analyses. Medication costs did not include costs of medications used during a long-term (>90 days) hospital stays because they are not registered in the FPR. Furthermore, we did not have information on patients using private healthcare services. However, as persons utilizing these services are not likely to use only private healthcare services in the care of T2D due to, for example, economic issues, this is not a big concern. Finally, we aimed to describe the incidence of complications and deaths by the trajectories. Thus, the results presented do not imply causal relationships since trajectories and incidence of the events were measured simultaneously.

Data-driven HbA1c trajectories among incident T2D patients may be utilized as a first step towards individualized care. Repeated measurements of HbA1c are available from EHRs enabling monitoring of patients' HbA1c levels during a longer period than one measurement point that is often used as a basis for treatment decisions. With the results of this study, clinicians are provided with more information to detect and treat patients whose diabetes care may need more attention. In addition, as patients have a central role in managing their diabetes care, demonstrating their future scenarios including risk of complications, death and healthcare costs may motivate them to engage more intensively to diabetes care. These aspects enable more personalized treatment of T2D patients. Further, this study shows the importance of proper diabetes care and glycaemic management for the society via the association between glycaemic control and medication costs. According to the results, special effort should be allocated to early detection of glycaemic control of late diagnosed T2D patients with high HbA1c values at the time of diagnosis and those with rapid deterioration in treatment balance to decrease the incidence of complications and medication costs. In future, this study should be replicated with a larger sample of T2D patients and longer follow-up to examine causal relationships.

## Supporting information

**S1 Table. Definitions of micro- and macrovascular complications and numbers of outcome events by type of analysis (first event occurrence after type 2 diabetes diagnosis).**
(PDF)

**S2 Table. Definitions of concordant and discordant diseases at baseline.**
(PDF)

**S3 Table. Unadjusted latent class growth analyses for HbA1c values measured annually since T2D diagnosis among patients diagnosed in 2011 or 2012 (n = 1540).**
(PDF)

**S4 Table. Antidiabetic medication use (%) by the trajectories after the T2D diagnosis.**
(PDF)

**S5 Table. Associations between the estimated HbA1c trajectories and incidence of deaths and diabetes complications.**
(PDF)

**S6 Table. Average accumulated 4-year per-person healthcare costs (euros, €) by HbA1c trajectories in 2014–2017.** All the costs are adjusted for censoring due to death utilizing Kaplan-Meier sample average estimator.
(PDF)

## Author Contributions

**Conceptualization:** Piia Lavikainen, Emma Aarnio, Kari Jalkanen, Tiina Laatikainen, Janne Martikainen.

**Formal analysis:** Piia Lavikainen, Miika Linna.

**Funding acquisition:** Tiina Laatikainen.

**Methodology:** Piia Lavikainen.

**Project administration:** Tiina Laatikainen, Janne Martikainen.

**Resources:** Miika Linna, Tiina Laatikainen.

**Software:** Piia Lavikainen.

**Supervision:** Hilkka Tirkkonen, Päivi Rautiainen, Tiina Laatikainen, Janne Martikainen.

**Visualization:** Piia Lavikainen.

**Writing – original draft:** Piia Lavikainen.

**Writing – review & editing:** Piia Lavikainen, Emma Aarnio, Miika Linna, Kari Jalkanen, Hilkka Tirkkonen, Päivi Rautiainen, Tiina Laatikainen, Janne Martikainen.

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
