## [Decision Letter · Decision Letter 0]

1 Apr 2022

PONE-D-22-05229Data-driven long-term glycemic control clusters and their associated health and economic outcomes in patients with incident type 2 diabetesPLOS ONE

Dear Dr. Lavikainen,

Thank you for submitting your manuscript to PLOS ONE. After careful consideration, we feel that it has merit but does not fully meet PLOS ONE’s publication criteria as it currently stands. Therefore, we invite you to submit a revised version of the manuscript that addresses the points raised during the review process.

We look forward to receiving your revised manuscript.

Kind regards,

Dylan A Mordaunt

Academic Editor

PLOS ONE

Journal Requirements:

[I have read the journal's policy and the authors of this manuscript have the following competing interests: JM is a founding partner of ESiOR Oy and a board member of Siltana Oy. These companies were not involved in carrying out this research. PL, EA, ML, KJ, HT, PR and TL declare no conflicts of interest.]

Additional Editor Comments:

Thank you for your submission. One of our reviewers has recommended rejection, however the rational is due to a perceived lack of novelty. PLoS One criteria for publication does not include novelty, but rather originality of work. The work does appear to be original. With regards to novelty, the reviewer makes suggestions for improvement that I think the authors should consider. It's also worth considering whether this is novel in the context of the Finish health system and society, since economic outcomes in particular would be health system and society specific. This is the reason I have not receommended rejecting the manuscript. My major area of comment is that any perceived novelty can and should be made clearer, though I've opted to call this a minor rather than major revision.

With specific regards to the criteria for publication:

1. The study appears to present the results of original research.

2. Results do not appear to have been reported elsewhere.

3. Experiments, statistics, and other analyses are performed to a high technical standard and are described in sufficient detail. One reviewer has suggested the authors consider further stratification. I also suggest considering making it clear (e.g. in the title) that this is within the context of Finland, though there may well be generalisable elements or insights.

4. Conclusions are presented in an appropriate fashion and are supported by the data.

5. The article is presented in an intelligible fashion and is written in standard English. Some minor suggestions are made by one of the reviewers, though to a technical English reader, this could affect the perceived credibility of the work since the English is otherwise of a good standard.

6. The research meets all applicable standards for the ethics of experimentation and research integrity.

7. The article could be refined against an appropriate observational study checklist from the Equator Network, such as one of the STROBE checklists. This adds rigour to the reporting. The checklist should be attached to the resubmission as it also adds value to the submission in terms of later meta-research on the resulting publication.

Reviewers' comments:

Reviewer's Responses to Questions

**Comments to the Author**

1. Is the manuscript technically sound, and do the data support the conclusions?

Reviewer #1: Yes

Reviewer #2: Yes

2. Has the statistical analysis been performed appropriately and rigorously? 

Reviewer #1: Yes

Reviewer #2: I Don't Know

3. Have the authors made all data underlying the findings in their manuscript fully available?

Reviewer #1: Yes

Reviewer #2: Yes

4. Is the manuscript presented in an intelligible fashion and written in standard English?

Reviewer #1: Yes

Reviewer #2: Yes

5. Review Comments to the Author

Reviewer #1: a useful work which confirms already accepted knowledge regarding the nature and natural history of diabetes, would have been add on to currently known data if there was any stratifications of the drug used and if there was any impact of the class of drugs especially new classes with presumed additive benefits on cardiovascular outcomes rather than just focusing on the progression of the disease

Reviewer #2: Piia Lavikainen et al have applied a clustering strategy to study patients with newly diagnosed type 2 diabetes and examined the association of these groups with heath outcomes including all-cause death and incident micro- and macrovascular complications.

4 groups were identified using this latent-class growth analysis clustering approach : “Stable, adequate” (large majority, 2/3 of the study population), “Slowly deteriorating” (second largest, one fourth of the population) and 2 smal lgroups “Rapidly deteriorating” glycaemic control (6.2%) and “Late diagnosed” patients (3.4%). Most of the interest is focused on these 2 last groups.

The paper is interesting but an effort should be made to facilitate direct access to salient clinical results

General comment

I see A1c not only as a biomarker of treatment efficacy but also as a possible biomarker indicative of severity of the disease as used in Ahlquist intial clustering work (Lancet DE). In addition the response of A1c cannot be seen as a universal response to drugs since diabetes drugs are limited to 8 large classes in patients with T2D (SU/glinides ; TZDs; Metformin; DPP4-Is ; SGLT2-Is; AG-Is; GLP1RA; insulin) although AG-Is do not appear in the current paper.

Table 1

The authors should present data on eGFR and not only serum creatinine values.

Do the authors have access to Fasting plasma glucose at diagnosis ? it would help to confront FPG and A1c.

Table 3 needs to be more directly self-comprehensive as the amount should correspond to yearly expenses (please clarify this)

I am wondering how to adjust on baseline characteristics which are different between the 4 selected groups, particularly age and sex, regarding the effect of each group on complications. This would be appreciated.

The discussion is interesting and could however be shortened.

6. PLOS authors have the option to publish the peer review history of their article (what does this mean?). If published, this will include your full peer review and any attached files.

Reviewer #1: No

Reviewer #2: No

---

## [Author Response · Author response to Decision Letter 0]

25 Apr 2022

Reviewers' comments:

Reviewer's Responses to Questions

Comments to the Author

1. Is the manuscript technically sound, and do the data support the conclusions?

Reviewer #1: Yes

Reviewer #2: Yes

2. Has the statistical analysis been performed appropriately and rigorously?

Reviewer #1: Yes

Reviewer #2: I Don't Know

3. Have the authors made all data underlying the findings in their manuscript fully available?

Reviewer #1: Yes

Reviewer #2: Yes

4. Is the manuscript presented in an intelligible fashion and written in standard English?

Reviewer #1: Yes

Reviewer #2: Yes

5. Review Comments to the Author

Reviewer #1: a useful work which confirms already accepted knowledge regarding the nature and natural history of diabetes, would have been add on to currently known data if there was any stratifications of the drug used and if there was any impact of the class of drugs especially new classes with presumed additive benefits on cardiovascular outcomes rather than just focusing on the progression of the disease

Our response: We thank the Reviewer for the comments. However, we would prefer not to conduct additional stratified analyses. First, diabetes medication use changes over time as is seen from figure 2 and table S4, which also show how medication use modifies HbA1c trajectories. Modeling such a time-varying factor that is also associated with HbA1c would be challenging in this study and we feel that stratified analyses would require different study design. Second, we have submitted another study where we have examined changes in diabetes medication use and its association with treatment outcomes.

Reviewer #2: Piia Lavikainen et al have applied a clustering strategy to study patients with newly diagnosed type 2 diabetes and examined the association of these groups with heath outcomes including all-cause death and incident micro- and macrovascular complications.

4 groups were identified using this latent-class growth analysis clustering approach : “Stable, adequate” (large majority, 2/3 of the study population), “Slowly deteriorating” (second largest, one fourth of the population) and 2 smal lgroups “Rapidly deteriorating” glycaemic control (6.2%) and “Late diagnosed” patients (3.4%). Most of the interest is focused on these 2 last groups.

The paper is interesting but an effort should be made to facilitate direct access to salient clinical results

General comment

I see A1c not only as a biomarker of treatment efficacy but also as a possible biomarker indicative of severity of the disease as used in Ahlquist intial clustering work (Lancet DE). In addition the response of A1c cannot be seen as a universal response to drugs since diabetes drugs are limited to 8 large classes in patients with T2D (SU/glinides ; TZDs; Metformin; DPP4-Is ; SGLT2-Is; AG-Is; GLP1RA; insulin) although AG-Is do not appear in the current paper.

Our response: We have added to the Discussion section on baseline clustering conducted by Ahlqvist et al. as follows: “To note, also clustering based on patient characteristics (age, HbA1c, BMI, homeostasis model assessment (HOMA2) estimates of β-cell function and insulin resistance) at the time of diagnosis in general diabetic population has been conducted[28] instead of longitudinal clustering based on HbA1c solely.”

We have also discussed about the HbA1c being possible indicator of disease severity as follows: “HbA1c may not reflect only treatment response but also the underlying severity of the disease[28]. Late diagnosed patients used most often insulin reflecting more severe disease, seen also as higher incidence of complications during the 7-year follow-up despite of reaching good glycaemic control 2–3 years after the diagnosis.”

AG-Is are not included in the paper as they are not used in Finland. AG-Is are not mentioned in the treatment guidelines of type 2 diabetes in Finland, and there seems to be no AG-I products on the market at the moment.

Table 1

The authors should present data on eGFR and not only serum creatinine values.

Do the authors have access to Fasting plasma glucose at diagnosis ? it would help to confront FPG and A1c.

Our response: We thank the Reviewer for the suggestion and have added eGFR and FPG to the table 1. FPG confronts the observed HbA1c values at baseline. Serum creatinine values were converted to eGFR utilizing the CKD-EPI (Chronic Kidney Disease Epidemiology Collaboration) equation (Levey et al. Ann Intern Med 2009;150:604-612) that accounts for age and gender. 

Table 3 needs to be more directly self-comprehensive as the amount should correspond to yearly expenses (please clarify this)

Our response: We thank the Reviewer for commenting on this. We have clarified that the table presents average accumulated 4-year per-person costs over 2014-2017 for the stable cluster and additional costs in other clusters compared with the stable one. We would prefer presenting the accumulated costs along the HbA1c trajectories rather than presenting average yearly costs although they are often reported in cost-effectiveness analyses. 

I am wondering how to adjust on baseline characteristics which are different between the 4 selected groups, particularly age and sex, regarding the effect of each group on complications. This would be appreciated.

Our response: We thank the Reviewer for noticing this. Individual level Cox regression models were used to examine adjusted associations of the 4 groups with complications. The models were adjusted for age, sex, concordant and discordant diseases, and microvascular and macrovascular comorbidities of a patient at baseline. Unfortunately, we had not mentioned this in the paragraph of Statistical analyses. Therefore, we have refined the text as follows: “Models were adjusted for age, sex, concordant and discordant diseases, and microvascular and macrovascular comorbidities at baseline.”

The discussion is interesting and could however be shortened.

Our response: We have shortened the discussion by removing, for example, detailed description of HbA1c clusters identified in previous studies.

6. PLOS authors have the option to publish the peer review history of their article (what does this mean?). If published, this will include your full peer review and any attached files.

Do you want your identity to be public for this peer review? For information about this choice, including consent withdrawal, please see our Privacy Policy.

Reviewer #1: No

Reviewer #2: No

---

## [Editor Report · Decision Letter 1]

18 May 2022

Data-driven long-term glycaemic control trajectories and their associated health and economic outcomes in Finnish patients with incident type 2 diabetes

PONE-D-22-05229R1

Dear Dr. Lavikainen,

We’re pleased to inform you that your manuscript has been judged scientifically suitable for publication and will be formally accepted for publication once it meets all outstanding technical requirements.

Kind regards,

Dylan A Mordaunt, MD, MPH, FRACP

Academic Editor

PLOS ONE

Additional Editor Comments (optional):

Thank you for your resubmission. This now meets the criteria for publication.
---

## [Editor Report · Acceptance letter]

23 May 2022

PONE-D-22-05229R1 

Data-driven long-term glycaemic control trajectories and their associated health and economic outcomes in Finnish patients with incident type 2 diabetes 

Dear Dr. Lavikainen:

I'm pleased to inform you that your manuscript has been deemed suitable for publication in PLOS ONE. Congratulations! Your manuscript is now with our production department. 

Kind regards, 

on behalf of

Associate Professor Dylan A Mordaunt 

Academic Editor

PLOS ONE